# Seasonal features of geomagnetic activity: a study on the solar activity dependence

Adriane Marques de Souza Franco[1], Rajkumar Hajra[2], Ezequiel Echer[1], and Mauricio José Alves Bolzan[3]

[1]Instituto Nacional de Pesquisas Espaciais (INPE), São José dos Campos, Brazil
[2]Indian Institute of Technology Indore, Simrol, Indore 453552, India
[3]Federal University of Jatai, Jatai, Brazil

**Correspondence:** Adriane Marques de Souza Franco (adrianemarquesds@gmail.com)

**Abstract.** Seasonal features of geomagnetic activity and their solar wind-interplanetary drivers are studied using more than 5 solar cycles of geomagnetic activity and solar wind observations. This study involves a total of 1296 geomagnetic storms of varying intensity identified using the Dst index from January 1963 to December 2019, a total of 75863 substorms identified from the SML index from January 1976 to December 2019, and a total of 145 high-intensity long-duration continuous auroral

electrojet (AE) activity (HILDCAA) events identified using the AE index from January 1975 to December 2017. The occurrence rates of the substorms, geomagnetic storms, including moderate ($-50$ nT $\geq$ Dst $> -100$ nT) and intense ($-100$ nT $\geq$ Dst $> -250$ nT), exhibit a significant semi-annual variation (periodicity $\sim 6$ months), while the super storms (Dst $\leq -250$ nT) and HILDCAAs do not exhibit any clear seasonal feature. The geomagnetic activity indices Dst and ap exhibit a semi-annual variation, while AE exhibits an annual variation (periodicity $\sim 1$ year). The annual and semi-annual variations are attributed

to the annual variation of the solar wind speed $V_{sw}$, and the semi-annual variation of the coupling function $VB_s$ (where $V = V_{sw}$, and $B_s$ is the southward component of the interplanetary magnetic field), respectively. We present a detailed analysis of the annual and semi-annual variations, and their dependencies on the solar activity cycles separated as the odd, even, weak and strong solar cycles.

## 1   Introduction

Solar wind-magnetosphere energy coupling causes disturbances in the magnetosphere of the Earth (e.g., Dungey, 1961; Axford and Hines, 1961; Tsurutani et al., 1992; Gonzalez et al., 1994; Tsurutani et al., 2020). Depending on the strength, duration and efficiency of the coupling, resultant geomagnetic disturbances (von Humboldt, 1808) can be classified as magnetic storms, substorms and high-intensity long-duration continuous auroral electrojet (AE) activities (HILDCAAs) (see Gonzalez et al., 1994; Hajra et al., 2020; Hajra, 2021a). In general, magnetic storms represent global-scale disturbances caused by enhance-

ments in (westward) ring current flowing at $\sim 2 - 7$ Earth radii ($R_\oplus$) in the magnetic equatorial plane of the Earth (Gonzalez et al., 1994; Lakhina and Tsurutani, 2018, and references therein). Storm duration spans a few hours to several days. In fact, while the storm main phase lasts typically for $\sim 10 - 15$ hours, the recovery phase can continue much longer, from hours to several days (Gonzalez et al., 1994). Substorms (Akasofu, 1964) are shorter-scale, a few minutes to a few hours, disturbances

in the nightside magnetosphere (magnetotail) resulting in precipitations of $\sim 10-100$ keV electrons and protons in the auroral ionosphere (e.g., Meng et al., 1979; Thorne et al., 2010; Tsurutani et al., 2019, and references therein). Intense auroral substorms continuing for a few days without occurrence of any major magnetic storms have been called HILDCAAs (Tsurutani and Gonzalez, 1987; Hajra et al., 2013) to distinguish them from nominal substorms and major magnetic storms (Tsurutani et al., 2004; Guarnieri, 2006).

It is important to note that from the physical point of view, substorms and HILDCAAs are two different types of geomagnetic activity. While substorms may occur during HILDCAAs, they represent different magnetosphere/ionosphere processes (Tsurutani et al., 2004; Guarnieri, 2005, 2006). For example, HILDCAAs are associated with Alfvén wave trains carried by solar wind high-speed ($\sim$550–850 km s$^{-1}$) streams (HSSs) emanated from solar coronal holes (Tsurutani and Gonzalez, 1987; Hajra et al., 2013). The intermittent magnetic reconnection between the Alfvén wave southward component and geomagnetic field results in intermittent increases in auroral activity during HILDCAAs. Substorms, on the other hand, are associated with solar wind energy loading in the magnetotail caused by magnetic reconnection (Tsurutani and Meng, 1972), and subsequent explosive release of the energy in form of energetic particles and strong plasma flows (e.g., Akasofu, 1964, 2017; Rostoker, 2002; Nykyri et al., 2019, and references therein). These are not essentially associated with HSSs. Thus, for good reason, the term "substorm" was avoided in the definition of HILDCAAs by Tsurutani and Gonzalez (1987). Later, Hajra et al. (2014b, 2015a, b) have shown that HILDCAAs take an important role in the acceleration of relativistic ($\sim$ MeV) electrons in the outer radiation belt of the Earth. This feature further distinguishes the HILDCAAs from nominal substorms.

Geomagnetic activity, in general, is known to be highly variable, modulated by several solar-terrestrial features. The solar/interplanetary sources of the variability include the $\sim 27$-day solar rotation (Bartels, 1932, 1934; Newton and Nunn, 1951), the $\sim 11$-year solar activity cycle (Schwabe, 1844), the electromagnetic and corpuscular radiations from the Sun, several plasma emission phenomena, heliospheric current region, etc. On the other hand, the Earth's translational movement (solstices), the inter-hemispheric symmetry (equinoxes), and the observational frame of reference or the coordinate system (Russell, 1971) can also largely impact the geomagnetic activity variation.

One of the earliest-reported features of the geomagnetic activity is the semi-annual variation, that is, more frequent occurrences and higher strength during equinoxes and lesser occurrences and weaker strength during solstices (e.g., Broun, 1848; Sabine, 1852). The semi-annual variation is reported in the occurrence rates and intensities of the magnetic storms (e.g., Cliver et al., 2000, 2004; Le Mouël et al., 2004; Cnossen and Richmond, 2012; Danilov et al., 2013; McPherron and Chu, 2018; Lockwood et al., 2020), and in the Earth's radiation belt electron variations (e.g., Baker et al., 1999; Li et al., 2001; Kanekal et al., 2010; Katsavrias et al., 2021). This is generally explained in the context of the Earth's position in the heliosphere (known as the "axial effect"; Cortie, 1912), relative angle of solar wind incidence with respect to Earth's rotation axis (the "equinoctial effect"; Boller and Stolov, 1970), and geometrical controls of interplanetary magnetic fields (the "Russell–McPherron effect"; Russell and McPherron, 1973). See Lockwood et al. (2020) for an excellent discussion of the mechanisms. While both the equinoctial and the Russell–McPherron effects are shown to be responsible for the semi-annual variation in the geomagnetic indices (e.g., Cliver et al., 2000; O'Brien and McPherron, 2002), the semi-annual variation in the relativistic electron fluxes of the outer belt is mainly attributed to the Russell–McPherron effect (e.g., Kanekal et al., 2010; Katsavrias et al., 2021).

**Table 1.** Details of the geomagnetic activity events under present study

| Events | Number of events | Periods of observation | Geomagnetic indices | Sources of events |
|---|---|---|---|---|
| Substorms | 75863 | January 1976–December 2019 | SML | https://supermag.jhuapl.edu/ |
| HILDCAAs | 145 | January 1975–December 2017 | AE, Dst | (Hajra et al., 2021) |
| Geomagnetic storms | 1296 | January 1963–December 2019 | Dst | (Hajra et al., 2021) |

However, the semi-annual variation in general was questioned by the work of Mursula et al. (2011) reporting solstice maxima in substorm frequency and duration, and substorm amplitude and global geomagnetic activity peaks alternating between spring and fall in $\sim 11$ years. While solstice maxima were attributed to auroral ionospheric conductivity changes (Wang and Lühr, 2007; Tanskanen et al., 2011), the alternating equinoctial maxima were associated to asymmetric solar wind distribution in solar hemispheres (Mursula and Zieger, 2001; Mursula et al., 2002). In addition, several recent studies have reported lack of any seasonal dependence of substorms (Hajra et al., 2016), HILDCAAs (Hajra et al., 2013, 2014a), and in the radiation belts (Hajra, 2021b).

In the present work, for the first time, we will explore a long-term database of substorms, HILDCAAs, and magnetic storms of varying intensity along with different geomagnetic indices to study the seasonal features of geomagnetic disturbances. The main aim is to identify and characterize the seasonal features of geomagnetic disturbances of different types and intensities. In addition, we will study their solar activity dependencies, if any.

## 2 Database and Methods

Details of the geomagnetic events studied in this work are summarized in Table 1. Auroral substorms are identified by intensification in the auroral ionospheric (westward) electrojet currents. In the present work, we will use the substorm list available at the SuperMAG website (https://supermag.jhuapl.edu/, Newell and Gjerloev, 2011; Gjerloev, 2012). The substorm expansion phase onsets were identified from the SML index which is the SuperMAG equivalent of the westward auroral electrojet index AL (see the cited references for details). The present work involves a total of 75863 substorms identified from January 1976 to December 2019 (Table 1).

We will use the geomagnetic storm and HILDCAA database prepared by Hajra et al. (2021) for the present work. It is an updated version of the lists presented in Echer et al. (2011), Hajra et al. (2013), and Rawat et al. (2018). Geomagnetic storm onset, main phase, peak strength, recovery phase, and storm end are determined by the variations of the Dst index (Sugiura, 1964). Based on Gonzalez et al. (1994) definition, intervals with the Dst minimum $\leq -50$ nT are identified as magnetic storms. From January 1963 to December 2019, 1296 magnetic storms were identified (Table 1). Geomagnetic storms with the Dst minimum values between $-50$ nT and $-100$ nT are classified as the "moderate storms", between $-100$ nT and $-250$ nT as the "intense storms", and those with the Dst minima lower than $-250$ nT as the "super storms". Among all storms studied here, 75% are moderate, 23% are intense, and only 2% are super storms.

**Table 2.** Details of the solar cycles under present study

| SC no. | SC start date (year-month) | SC peak date (year-month) | SC peak $F_{10.7}$ | SC end date (year-month) |
|--------|------------|-----------|------------------|------------|
| SC20 | 1964-10 | 1968-11 | 156 | 1976-02 |
| SC21 | 1976-03 | 1979-12 | 203 | 1986-08 |
| SC22 | 1986-09 | 1989-11 | 213 | 1996-07 |
| SC23 | 1996-08 | 2001-11 | 181 | 2008-11 |
| SC24 | 2008-12 | 2014-04 | 146 | 2019-12 |

The HILDCAA events are identified based on four criteria suggested by Tsurutani and Gonzalez (1987). The criteria are: (1) the AE index should reach an intensity equal to or greater than 1000 nT at some point during the event (the high-intensity criterion), (2) the event must last at least 2 days (the long-duration criterion), (3) the AE index should not fall below 200 nT for more than 2 h at a time (the continuity criterion), and (4) the auroral activity must occur outside the main phase of a geomagnetic storm or during a non-storm condition (Dst $> -50$ nT). Present work involves a total of 145 HILDCAA events identified during January 1975 through December 2017 (Table 1).

The geomagnetic indices, namely the ring current index Dst, the global-scale geomagnetic activity index ap, and the auroral ionospheric current related index AE, are used to provide a quantitative measure of the activity level of the terrestrial magnetosphere (Rostoker, 1972). In addition, solar wind parameters are used to study the energy dissipation in the magnetosphere. The $D_{500}$ parameter is defined as the percentage of days with the peak solar wind speed $V_{sw}$ equal or higher than 500 km s$^{-1}$ in each month of a year. We estimated the solar wind electric field $VB_s$, which is an important solar wind-magnetosphere coupling function (Burton et al., 1975; Tsurutani et al., 1992; Finch et al., 2008). As $VB_s$ involves both the solar wind velocity $V_{sw}$ (for $V$) and the southward component of the interplanetary magnetic field (IMF) $B_s$, the latter being important for magnetic reconnection, $VB_s$ is also called the reconnection electric field. The Akasofu-$\epsilon$ coupling function (Perreault and Akasofu, 1978), expressed as: $V_{sw}B_0^2 sin^4(\theta/2)R_{CF}^2$, was also estimated in this work as a proxy for the magnetospheric energy input rate. Here $B_0$ represents the magnitude of the IMF, $\theta$ is the IMF orientation clock angle, and $R_{CF}$ is the Chapman-Ferraro magnetopause distance (Chapman and Ferraro, 1931).

The 10.7 cm solar flux ($F_{10.7}$) is shown to be a good indicator of the solar activity (e.g., Tapping, 1987). Thus, the $\sim$11-year solar cycles (Schwabe, 1844) are studied using the monthly mean $F_{10.7}$ solar flux variation. The starting, peak and end dates along with the peak $F_{10.7}$ flux of each solar cycle are listed in Table 2. The $F_{10.7}$ fluxes are given in the solar flux unit (sfu), where 1 sfu = $10^{-22}$ W m$^{-2}$ Hz$^{-1}$. Based on the $F_{10.7}$ peaks, cycles SC20 and SC24 can be classified as the "weak cycles" (average $F_{10.7}$ peak $\sim$ 151 sfu), and SC19, SC21, SC22 and SC23 as the "strong cycles" (average $F_{10.7}$ peak $\sim$ 207 sfu). It can be mentioned that SC24 is the weakest cycle in the space exploration era (after 1957). A detailed study on the solar and geomagnetic characteristics of this cycle is presented in Hajra (2021c). The solar cycles are also grouped into the "even" (SC20, SC22, SC24) and the "odd" (SC19, SC21, SC23) cycles in this work. Previous studies have reported significant differences

between the even and odd cycle amplitudes (e.g., Waldmeier, 1934; Gnevyshev and Ohl, 1948; Wilson, 1988; Durney, 2000), and in their geomagnetic responses (e.g., Hajra et al., 2021; Owens et al., 2021).

We will apply the Lomb-Scargle periodogram analysis (Lomb, 1976; Scargle, 1982) to identify the significant periodicities in the geomagnetic event occurrences, the geomagnetic indices, and the solar wind-magnetosphere (coupling) parameters. It is a useful tool for detecting and characterizing periodic signals for unequally spaced data.

The geomagnetic indices are collected from the World Data Center for Geomagnetism, Kyoto, Japan (http://wdc.kugi. kyoto-u.ac.jp/). The monthly means of the solar wind/interplanetary data near the Earth's bow shock nose were obtained from NASA's OMNI database (http://omniweb.gsfc.nasa.gov/). The IMF vector components are in geocentric solar magnetospheric (GSM) coordinates, where the $x$-axis is directed towards the Sun and the $y$-axis is in the $\mathbf{\Omega} \times \hat{\mathbf{x}}/|\mathbf{\Omega} \times \hat{\mathbf{x}}|$ direction, where $\mathbf{\Omega}$ is aligned with the magnetic south pole axis of the Earth, and $\hat{\mathbf{x}}$ is the unit vector along the $x$-axis. The $z$-axis completes a right-

hand system. The $F_{10.7}$ solar fluxes are obtained from the Laboratory for Atmospheric and Space Physics (LASP) Interactive Solar Irradiance Data Center (https://lasp.colorado.edu/lisird/).

## 3 Results

### 3.1 Seasonal features

Figure 1 shows the variations of the monthly mean solar $F_{10.7}$ flux (Figure 1a), the monthly numbers of HILDCAAs and
125 substorms (Figure 1b), magnetic storms of varying intensity (Figure 1c), the monthly mean geomagnetic Dst (Figure 1d), ap (Figure 1e) and AE (Figure 1f) indices, the IMF magnitude $B_0$ (Figure 1g), the solar wind plasma speed $V_{sw}$ (Figure 1h), the percentage occurrences of $V_{sw} \geq 500$ km s$^{-1}$ ($D_{500}$, Figure 1i), and the energy coupling functions $VB_s$ (Figure 1j) and $\epsilon$ (Figure 1k) for the period from 1963 through 2019. While most of the data spans for more than five solar cycles, from the beginning of SC20 to the end of SC24, substorm and HILDCAA data are only available from SC21 onward. The $F_{10.7}$ solar
flux variation shows a clear $\sim$ 11-year solar activity cycle, with the minimum flux during the solar minimum, followed by flux increases during the ascending phase leading to the peak flux during the solar maximum, and flux decreases during the descending phase of the solar cycle (Figure 1a). In general, the substorm, HILDCAA and geomagnetic storm numbers, the geomagnetic indices and the solar wind parameter values exhibit an overall $\sim$ 11-year periodicity. Embedded in the large-scale $\sim$ 11-year variations, there are several short-term fluctuations in the data. Some of the latter may be associated with the annual
or semi-annual variations, which will be explored in detail in the following sections.

**Monthly superposed variations**

Figure 2 shows the monthly superposed values of all the parameters shown in Figure 1. The left panels show the numbers of geomagnetic events in each month divided by the number of years of observations (in the unit of number per year). The right panels show the monthly means of the geomagnetic and solar wind/interplanetary parameters for the entire interval of study.

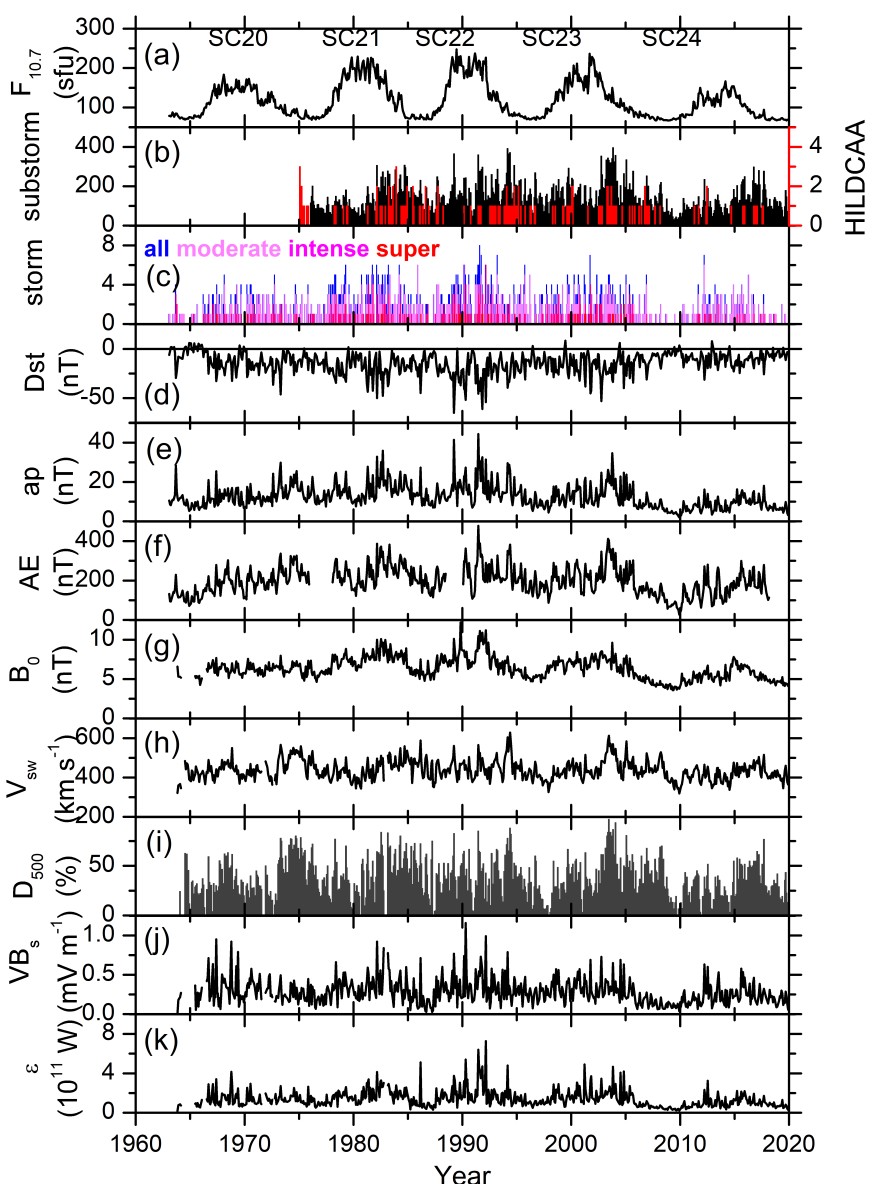

**Figure 1.** From top to bottom, the panels show (a) the monthly mean solar $F_{10.7}$ flux, monthly numbers of (b) substorms (black, legend on the left) and HILDCAAs (red, legend on the right), (c) geomagnetic storms of varying intensity, monthly mean (d) Dst, (e) ap, (f) AE, (g) IMF $B_0$, (h) $V_{sw}$, (i) percentage of days with daily peak $V_{sw} \geq 500$ km s$^{-1}$ ($D_{500}$), monthly mean (j) $VB_s$ and (k) Akasofu $\epsilon$-parameter, respectively during 1963 through 2020. Solar cycles from SC20 through SC24 are marked on the top panel.

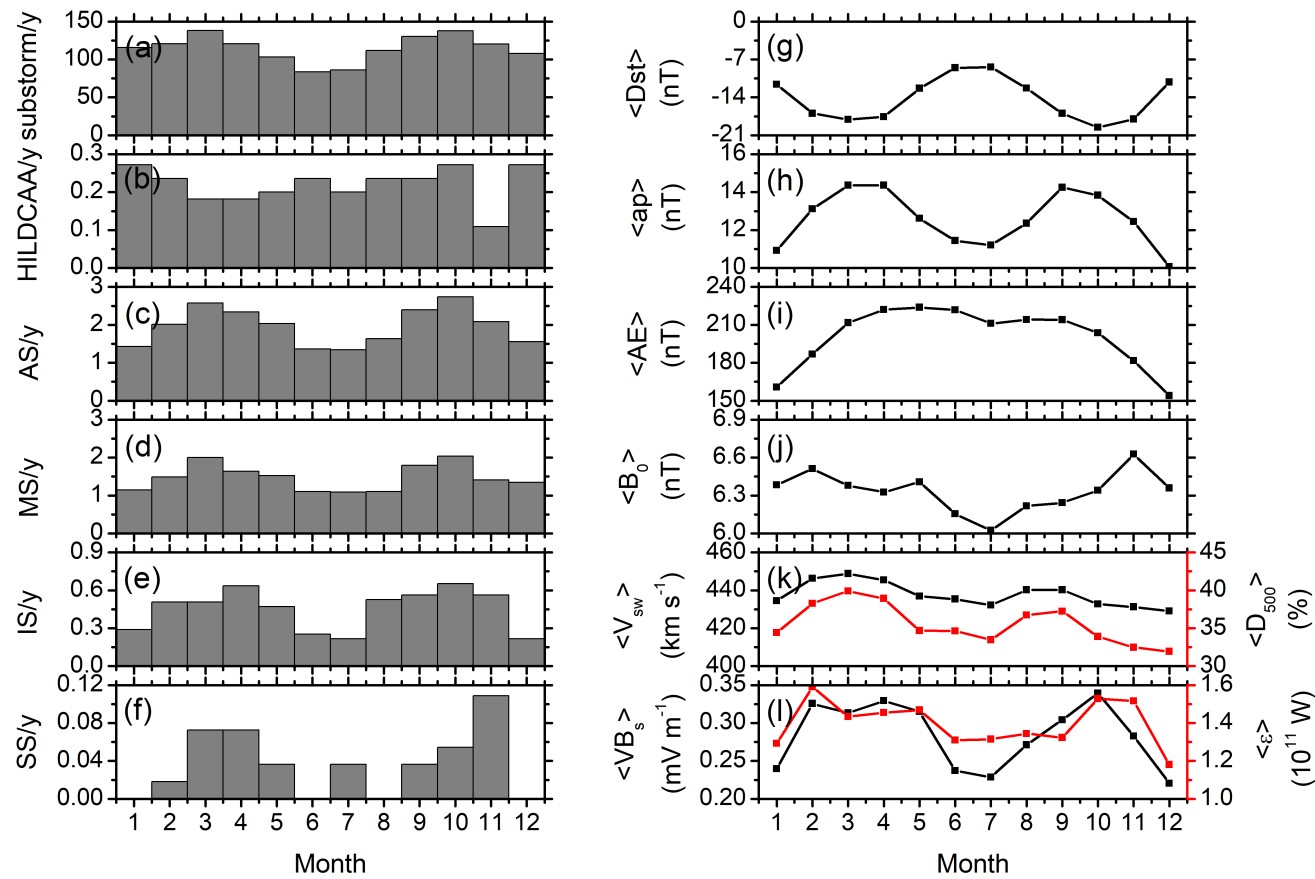

**Figure 2.** Monthly superposed variations. Left panels, from top to bottom, show the total numbers divided by years of observation of (a) substorms, (b) HILDCAAs, (c) all storms (AS), (d) moderate (MS), (e) intense (IS) and (f) super (SS) storms, respectively. Right panels, from top to bottom, show the monthly mean values of the geomagnetic (g) Dst, (h) ap and (i) AE indices, (j) IMF $B_0$, (k) $V_{sw}$ (black, legend on the left) and $D_{500}$ (red, legend on the right), and (l) $VB_s$ (black, legend on the left) and $\epsilon$-parameter (red, legend on the right), respectively.

The substorm occurrence rate (Figure 2a) clearly exhibits two peaks during the months of March and October, and a summer solstice minimum (during the month of June). HILDCAAs (Figure 2b) do not exhibit any clear seasonal feature, except a significant minimum in November. Geomagnetic storms, from moderate to intense (Figure 2d–e), exhibit a clear semi-annual variation. The spring equinoctial peak is recorded during March for the moderate storms, and during April for the intense storms, while the fall peak is recorded during October for both of them. The super storms (Figure 2f), with a very low occurrence rate, do not have any clear seasonal feature. As majority of the storms are of moderate intensity, storms of all intensity together (Figure 2c) exhibit a prominent semi-annual variation with two peaks during March and October.

The monthly mean intensities of the Dst (Figure 2g) and ap (Figure 2h) indices show a semi-annual variation. Both of them exhibit the spring peaks during March. While Dst has a fall minimum during October, ap exhibits a peak during September. On the other hand, the monthly mean AE index (Figure 2i) increases gradually from January, attains a peak around April, decreases with a much slower rate till September, after which the decrease rate is faster, and finally AE attains a minimum during December. Thus, the AE index shows an annual variation, different from the Dst and ap indices. This result is consistent with Katsavrias et al. (2016) who also reported an annual component in AE, and lack of any semi-annual component. As the AE index is based on geomagnetic observations made in the northern hemisphere, the asymmetric pole exposition to the solar radiation during the Earth's translational motion could contribute to this annual variation. The latter may modulate the AE current through the modulation of the ionospheric conductivity owing to the solar extreme ultraviolet (EUV) ionization.

It is worth mentioning that the AE index (Davis and Sugiura, 1966) includes an upper envelope (AU) and a lower envelope (AL) related to the largest (positive) and smallest (negative) magnetic deflections, respectively among the magnetometer stations used. The AU and AL components represent the strengths of the eastward and westward AE, respectively. Lockwood et al. (2020) showed that the semi-annual variation is indeed present in the AL index. As the auroral westward current represented by AL is associated with the substorm related energetic particle precipitation in the auroral ionosphere, the semi-annual variation in AL is consistent with the semi-annual variation exhibited by the substorms (present work). On the other hand, the eastward auroral current/AU is mainly contributed by the dayside ionospheric conductivity that exhibits a summer solstice maximum as suggested by Wang and Lühr (2007); Tanskanen et al. (2011).

Among the solar wind-magnetosphere coupling parameters, $VB_s$ (Figure 2l, legend on the left) exhibits a semi-annual variation, with larger average values during February-April months, another sharp peak during October and with a solstice minimum. For the monthly mean IMF $B_0$ (Figure 2j), a clear minimum can be noted during July, and $B_0$ increases gradually on both sides of July. No clear seasonal features can be inferred from the variations of the monthly mean $V_{sw}$ (Figure 2k, legend on the left), and Akasofu $\epsilon$-parameter (Figure 2l, legend on the right). However, $D_{500}$ (Figure 2k, legend on the right) exhibits two clear peaks around March and September, with prominently lower values during solstices.

**Periodogram analysis**

It should be noted that the seasonal features as described above (Figure 2) present an average scenario composed by superposition of several solar cycles. The seasonal features may vary from one solar cycle to the other. In Figure 3 we have performed the Lomb-Scargle periodogram analysis (Lomb, 1976; Scargle, 1982) of the above events and parameters. For this purpose, we

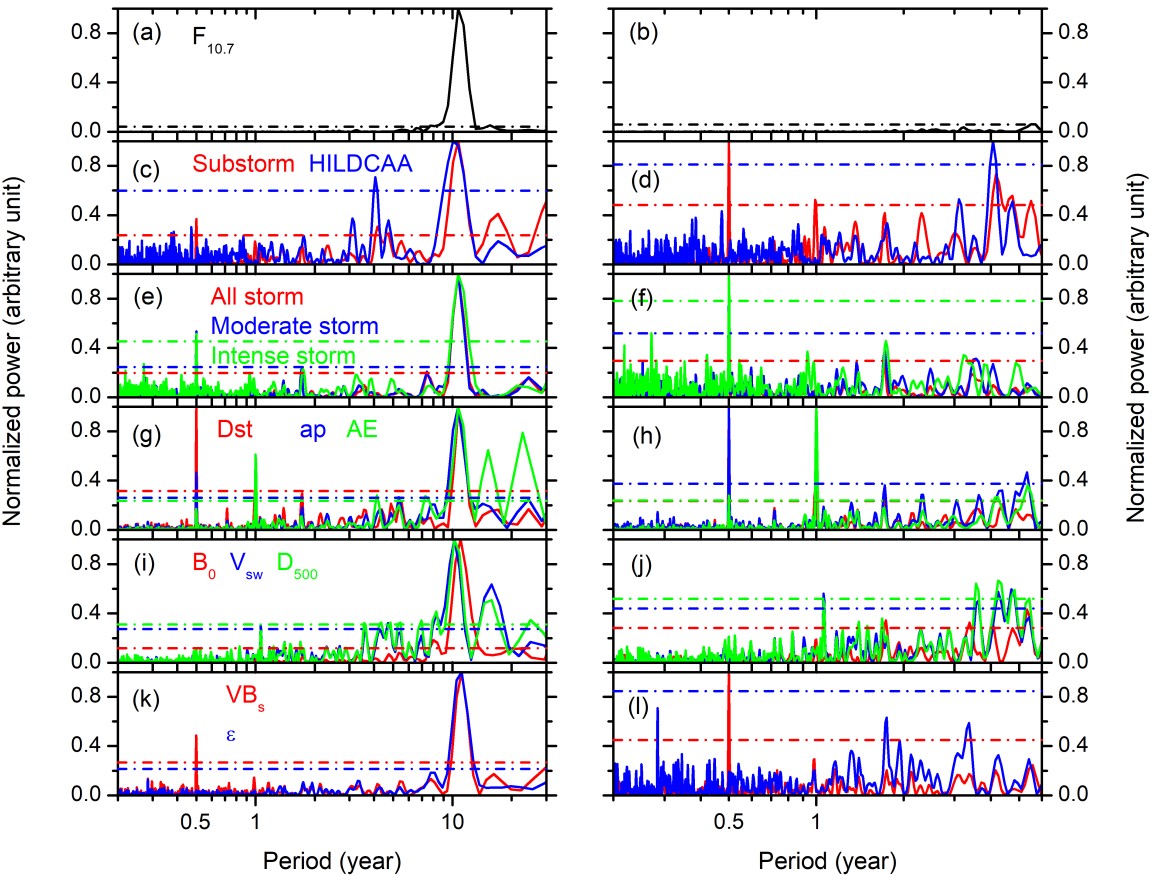

**Figure 3.** Lomb-Scargle periodograms. From top to bottom, the panels show the normalized power of periods for the monthly mean (a)–(b) solar $F_{10.7}$ flux, monthly numbers of (c)–(d) substorms and HILDCAAs, (e)–(f) all magnetic storms, moderate and intense storms, monthly mean (g)–(h) geomagnetic indices Dst, ap and AE, (i)–(j) solar wind parameters IMF $B_0$, $V_{sw}$ and $D_{500}$, (k)–(l) $VB_s$ and $\epsilon$-parameter, respectively. The left panel corresponds to periodograms of the original database without any filtering, while the right panel corresponds to periodograms after filtering out the 11-year periodicity from the database. Horizontal dash–dot lines in each panel indicate $> 95\%$ significance levels of the corresponding parameters shown by different colors. Note that the x-axes have different scaling for the left and right panels.

use the monthly means of $F_{10.7}$, Dst, ap, AE, $B_0$, $V_{sw}$, $D_{500}$, $VB_s$ and $\epsilon$, and the monthly numbers of substorms, HILDCAAs and magnetic storms of varying intensity. In the left panel of Figure 3, the periodograms are based on the original data of 1 month resolution, while the right panel shows the periodograms after filtering out the dominating $\sim 11$-year periodicity from the data. It can be noted that the filtering helps to better identify the shorter-scale periodicities in the time series.

**Table 3.** Significant (at the $> 95\%$ level) periods less than $\sim 11$ years obtained from the Lomb-Scargle periodogram analysis. Periods are ordered from higher power to lower.

| Events/parameters | Period (year) |
|---|---|
| geomagnetic activity: | |
| substorms | 0.5, 4.2 |
| HILDCAAs | 4.1 |
| all storms | 0.5 |
| moderate storms | 0.5 |
| intense storms | 0.5 |
| super storms | No |
| geomagnetic indices: | |
| Dst | 0.5 |
| ap | 0.5 |
| AE | 1.0 |
| solar wind parameters: | |
| $B_0$ | 8.0 |
| $V_{sw}$ | 8.3, 4.7, 1.1 |
| $D_{500}$ | 8.3, 7.0, 5.4, 4.8, 4.3, 3.6, 1.1 |
| $VB_s$ | 0.5 |
| $\epsilon$ | 8.1 |

As expected, the $F_{10.7}$ solar flux shows a prominent (at $> 95\%$ significance level) $\sim 11$-year periodicity (Figure 3a) and no shorter-scale variation (Figure 3b). A dominating $\sim 11$-year periodicity can also be observed in substorms, HILDCAAs (Figure 3c), magnetic storms of varying intensity (Figure 3e), the geomagnetic indices Dst, ap and AE (Figure 3g), and in the solar wind/interplanetary parameters IMF $B_0$, $V_{sw}$, $D_{500}$ (Figure 3i) and the solar wind-magnetosphere coupling functions $VB_s$ and $\epsilon$ (Figure 3k). However, we are interested in the annual or shorter-scale periodicities in the events and parameters. Thus, the Lomb-Scargle periodograms are also performed after filtering out this dominating $\sim 11$-year periodicity from the data. The same is shown in the right panel of Figure 3.

Table 3 lists the significant periodicities which are less than the $\sim 11$-year solar cycle period. As clear from Figure 3 and Table 3, substorms (Figure 3d), moderate and intense geomagnetic storms (Figure 3f) exhibit prominent semi-annual ($\sim 6$-month period) variation. However, the super storms do not exhibit any clear variation pattern (not shown). HILDCAAs (Figure 3d), on the other hand, exhibit a $\sim 4.1$-year periodicity, while no annual or lower-scale variation was recorded. However, it should be noted that very low monthly numbers of HILDCAAs and super storms during different years may introduce significant artifacts to the corresponding spectral/periodogram analysis. Thus, the results of the periodogram analysis for HILDCAAs and super storms cannot be fully trusted.

Both the ap and Dst indices exhibit a clear $\sim 6$-month periodicity (Figure 3h). However, the AE index exhibits an annual variation, but no semi-annual variation.

The solar wind/interplanetary and coupling functions exhibit more complex periodicity (lower than $\sim 11$-year). The IMF $B_0$ (Figure 3i) and $\epsilon$-parameter (Figure 3k) exhibit $\sim 8$-year periodicity, but no annual or lower-scale periodicity (Figures 3j and 3l). The solar wind $V_{sw}$ and $D_{500}$ (Figure 3j) exhibit several periodicities in the range of $\sim 4 - 8$ years and a significant annual variation (periodicity $\sim 1$ year). The coupling function $VB_s$ exhibits a prominent semi-annual variation (Figure 3l). The $V_{sw}$ periodicities detected in the present work are consistent with results reported previously (e.g., Valdés-Galicia et al., 1996; El-Borie, 2002; El-Borie et al., 2020; Hajra, 2021a; Hajra et al., 2021, and references therein). For example, El-Borie (2002) reported $\sim 9.6$-year periodicity in $V_{sw}$ arising from the coronal hole variations in the southern hemisphere of the Sun. El-Borie et al. (2020) discussed multiple $V_{sw}$ periodicities in the $1 - 2$-, $2 - 4$-, $4 - 8$- and $8 - 16$-year bands. Recently, Hajra et al. (2021) reported significant $V_{sw}$ periodicities of $\sim 3$, $\sim 4$, $\sim 10$ and $\sim 16$ years and discussed their important role in space climatology.

The results shown in Figure 3 and Table 3 are consistent with those in Figure 2. From the above analyses, the coupling function $VB_s$ which exhibits a $\sim 6$-month periodicity can be inferred as the driver of the semi-annual variations in substorms, moderate and intense storms, and in the geomagnetic indices Dst and ap. On the other hand, the $\sim 1$-year periodicity in $V_{sw}/D_{500}$ can be a source of the annual variation in the AE index. In addition, the $\sim 4.1$-year periodicity in HILDCAAs seems to be associated with the solar wind $V_{sw}$ variation in the same range. Detailed analyses of the events and/or parameters which exhibit the annual and/or semi-annual variations are shown in Section 3.2. For a detailed analysis of the longer-scale variations of the geomagnetic activity, the geomagnetic indices, and the solar wind-magnetosphere coupling, which is beyond the scope of this present work, we refer the read to Hajra et al. (2021).

## 3.2 Solar activity dependence

The solar cycle variations of the seasonal features described in Section 3.1 are explored in Figures 4 to 11. They show the variations of the substorms (Figure 4), the moderate (Figure 5) and intense (Figure 6) magnetic storms, the geomagnetic Dst (Figure 7), ap (Figure 8) and AE (Figure 9) indices, the solar wind plasma speed $V_{sw}$ (Figure 10), and the coupling function $VB_s$ (Figure 11). The format is identical for all these figures: for the geomagnetic events (the solar wind interplanetary parameters), panel (c) shows the year-month contour plot of the number of the events (the mean values) in each month of the observing years. The values of different colours are given in the legend at the bottom. Panel (d) shows the yearly mean $F_{10.7}$ solar flux. The solar minima are marked by the horizontal dash-dot lines in the bottom panels (c–d). Panel (b) shows the monthly numbers of the events per a year of observation (the monthly mean values of the parameters) during each solar cycle, while panel (a) shows the same during groups of the even, odd, strong, weak and all solar cycles.

Table 4 lists a "seasonal modulation" parameter defined as the difference between the equinoctial maximum and the solstice minimum expressed as the percentage of the yearly mean value for the events and parameters exhibiting the semi-annual variation. The modulation parameter can be taken as a measure of the seasonal/semi-annual variability. Larger the value of the parameter, stronger the semi-annual variability. Large variation in the seasonal modulation can be noted from the table. For

**Table 4.** Seasonal modulation (%) between the equinoctial maximum and the solstice minimum for the events and the parameters with the semi-annual variation during the weak and strong solar cycles, and the odd and even solar cycles (defined in Section 2).

| Events/parameters | Weak solar cycle | Strong solar cycle | Odd solar cycle | Even solar cycle |
|---|---|---|---|---|
| substorms | 55 | 46 | 49 | 66 |
| all storms | 85 | 76 | 76 | 78 |
| moderate storms | 92 | 73 | 68 | 77 |
| intense storms | 92 | 100 | 133 | 105 |
| Dst | 67 | 85 | 96 | 79 |
| ap | 40 | 37 | 38 | 46 |
| $VB_s$ | 54 | 57 | 53 | 40 |

substorms, all storms, moderate storms and the ap index, seasonal modulations are larger during the weak cycles (even cycles) than the strong cycles (odd cycles). However, the modulations are larger during the strong cycles (odd cycles) than the weak cycles (even cycles) for the intense storms, the Dst index and the coupling function $VB_s$. The explanation is not known at present. However, it is interesting to note that the intense storms (and thus the strong Dst associated with intense $VB_s$) are

mainly driven by the interplanetary coronal mass ejections (ICMEs). On the other hand, the moderate storms, substorms, and the ap index variations are associated with both ICMEs, and the corotating interaction regions (CIRs) between the slow streams and HSSs (e.g., Tsurutani and Gonzalez, 1987; Tsurutani et al., 1988; Gosling et al., 1990; Richardson et al., 2002; Echer et al., 2008; Hajra et al., 2013; Souza et al., 2016; Mendes et al., 2017; Marques de Souza et al., 2018; Tsurutani et al., 2019, and references therein). The strong cycles are expected to be characterized by more solar transient events like ICMEs than during

the weak cycles. However, recent studies show lower numbers and reduced geoeffectivenesses of both CIRs and ICMEs during the weak cycles than during the strong cycles (e.g., Scolini et al., 2018; Grandin et al., 2019; Lamy et al., 2019; Nakagawa et al., 2019; Syed Ibrahim et al., 2019; Hajra, 2021c, and references therein). This calls for a further study to explain the above results.

**Substorms**

From Figure 4c it can be seen that in any solar cycle, the peak substorm occurrence rates are noted during the descending phase, followed by the occurrence minimum during the solar minimum to early ascending phase. From the complete four solar cycles (SC21–SC24) of the substorm observations, two prominent peaks can be noted in the years of 1994 and 2003, which are in the descending phases of SC22 and SC23, respectively.

On the seasonal basis, two peaks around the months of March and October can be observed from the year-month contour

plot (Figure 4c), which is also reflected in the monthly superposed plots (Figure 4a–b). However, this "semi-annual" variation exhibits a large asymmetry in amplitude and duration between the spring and fall equinoxes. For example, in the year 1994, the substorm occurrence peak during February-May is significantly larger than the occurrences during October. On the other hand, during 2003, while the occurrence peak is noted in November, comparable occurrences are clear almost during the entire year.

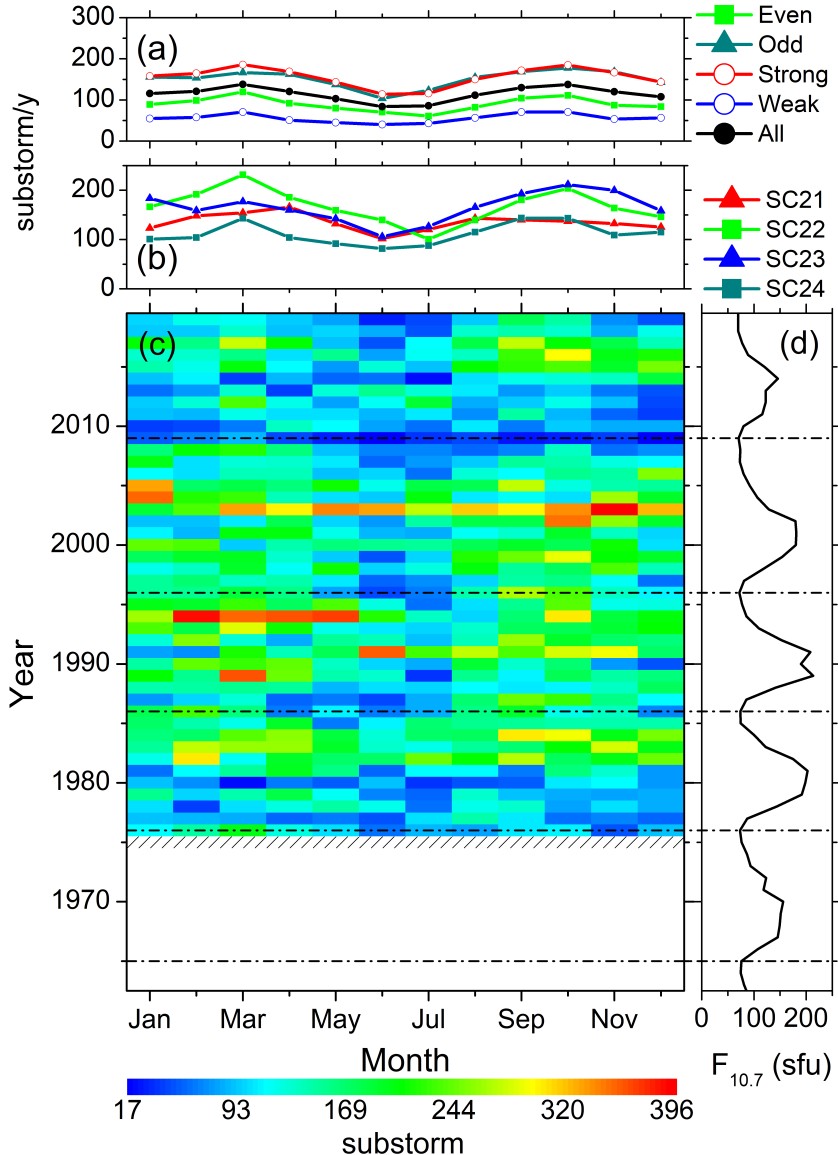

**Figure 4.** Substorms from 1976 through 2019. Panel (c) shows the year-month contour plot of the number of substorms in each month of the years 1976-2019. The values of different colours are given in the legend at the bottom. Data gaps are shown by crosses. Panel (d) shows the yearly mean $F_{10.7}$ solar flux. Panel (b) shows the monthly numbers of substorms per a year of observation during each solar cycles, and panel (a) shows the same during groups of the even, odd, strong, weak and all solar cycles. For details on the grouping of the solar cycles, see the text. The solar minima are marked by horizontal dash-dot lines.

When separated on the basis of the solar cycles (Figure 4a–b), the smallest numbers of events are observed during SC24. Interestingly, the spring occurrences are the strongest in SC22 and the fall occurrences are the strongest in SC23. Another noteworthy feature is that the occurrence rates during the even and weak solar cycles are lower than during the odd and strong cycles, respectively. However, the seasonal modulation between the equinoctial maximum and the solstice minimum is comparable between the weak ($\sim 55\%$) and strong ($\sim 46\%$) cycles (Table 4).

## Geomagnetic storms

Variations of the moderate and intense geomagnetic storms are shown in Figures 5 and 6, respectively. From the year-month contour plots (Figures 5c and 6c), the moderate storms are found to peak around the descending phases, while the intense storms peak around the solar maximum. When the monthly variations of the storms are considered in each year, there is hardly any seasonal variation. However, when observations during several solar cycles are grouped together (Figures 5a and 6a), the semi-annual variation can be noted in the moderate storms. There is not much difference in moderate and intense storm occurrence rates between the odd and even cycles. However, the occurrence rates of the storms are slightly larger in the strong cycles compared to the weak ones, while the seasonal modulation between the equinoctial maximum and the solstice minimum during the strong and weak cycles is comparable (Table 4). Another noteworthy feature is the lowest occurrence of intense storms during SC24.

## Geomagnetic indices

Variations of the monthly mean geomagnetic indices are shown in Figures 7 (Dst), 8 (ap) and 9 (AE). In each solar cycle, the average Dst index exhibits the strongest negative excursions at and immediately after the solar maximum (Figure 7c–d). A clear correlation can be observed between the $F_{10.7}$ solar flux and the average Dst strength. The Dst negative excursions are stronger during the strong and odd cycles compared to the weak and even cycles, respectively (Figure 7a). In addition, the seasonal modulation between the equinox minimum to the solstice maximum is significantly higher in the strong cycles ($\sim 85\%$) compared to the weak cycles ($\sim 67\%$) (Table 4). During SC24, the overall Dst strength is the weakest and there is no prominent seasonal modulation.

Variation of the monthly mean ap index (Figure 8) is identical to the Dst index variation. However, the seasonal modulation is comparable between the strong ($\sim 37\%$) and weak ($\sim 40\%$) cycles for the ap index (Table 4).

Variation of the AE index (Figure 9) is significantly different than the variations of the Dst and ap indices. In a solar cycle, AE peaks around the descending phase (Figure 9c). On the yearly basis, the average AE values are enhanced from March/April to September/October. The summer solstice values are significantly higher compared to the winter solstice values. This indicates an annual variation, in agreement with the Lomb-Scargle periodogram analysis result (Figure 3h). There is no semi-annual variation. The average values during the strong and odd solar cycles are higher compared to the weak and even solar cycles, respectively (Figure 9a). SC24 exhibited the lowest values of AE compared to other solar cycles (Figure 9b).

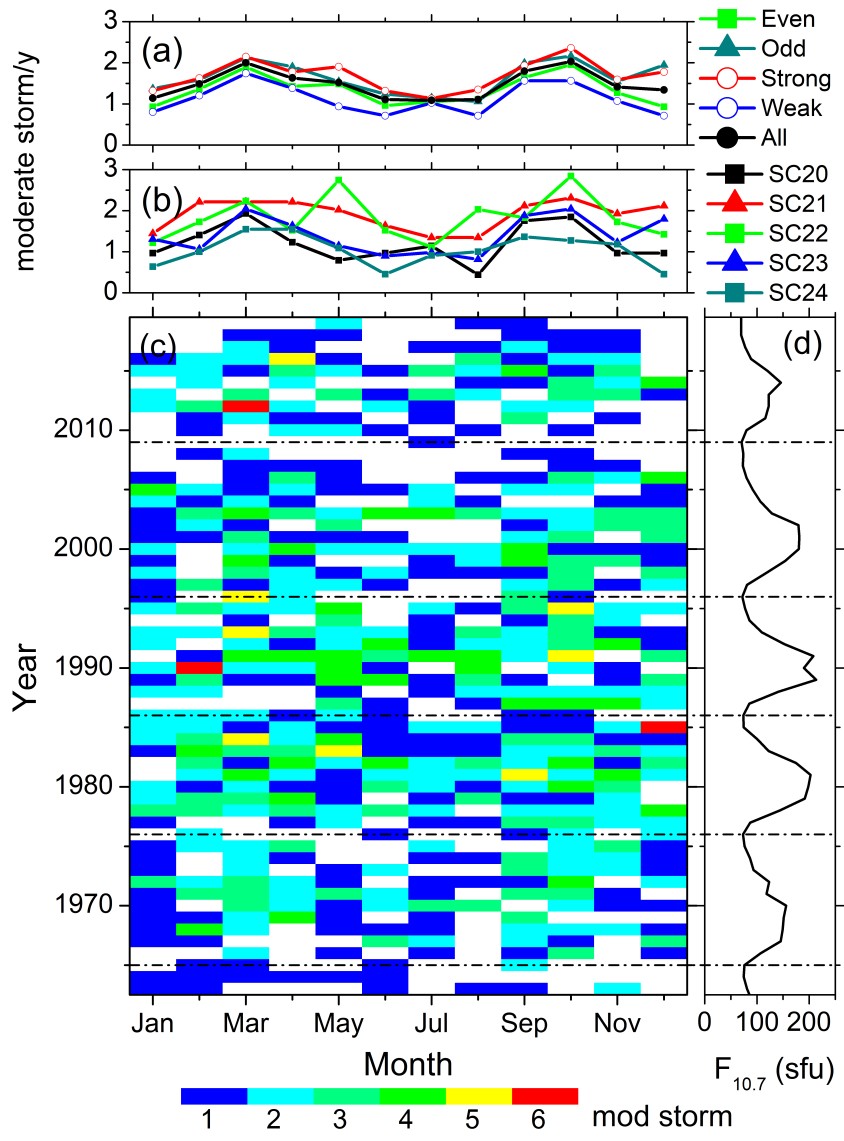

**Figure 5.** Moderate geomagnetic storms from 1963 through 2019. The panels are in the same format as in Figure 4.

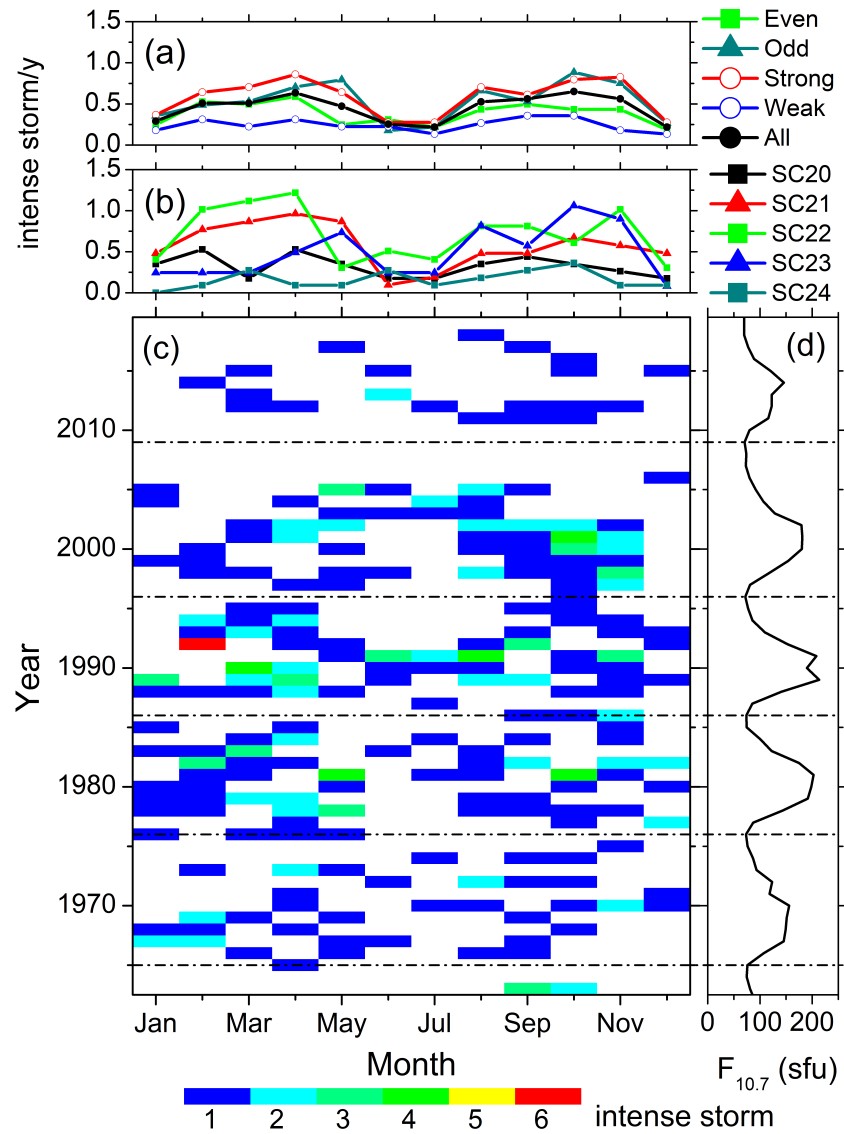

**Figure 6.** Intense geomagnetic storms from 1963 through 2019. The panels are in the same format as in Figure 4.

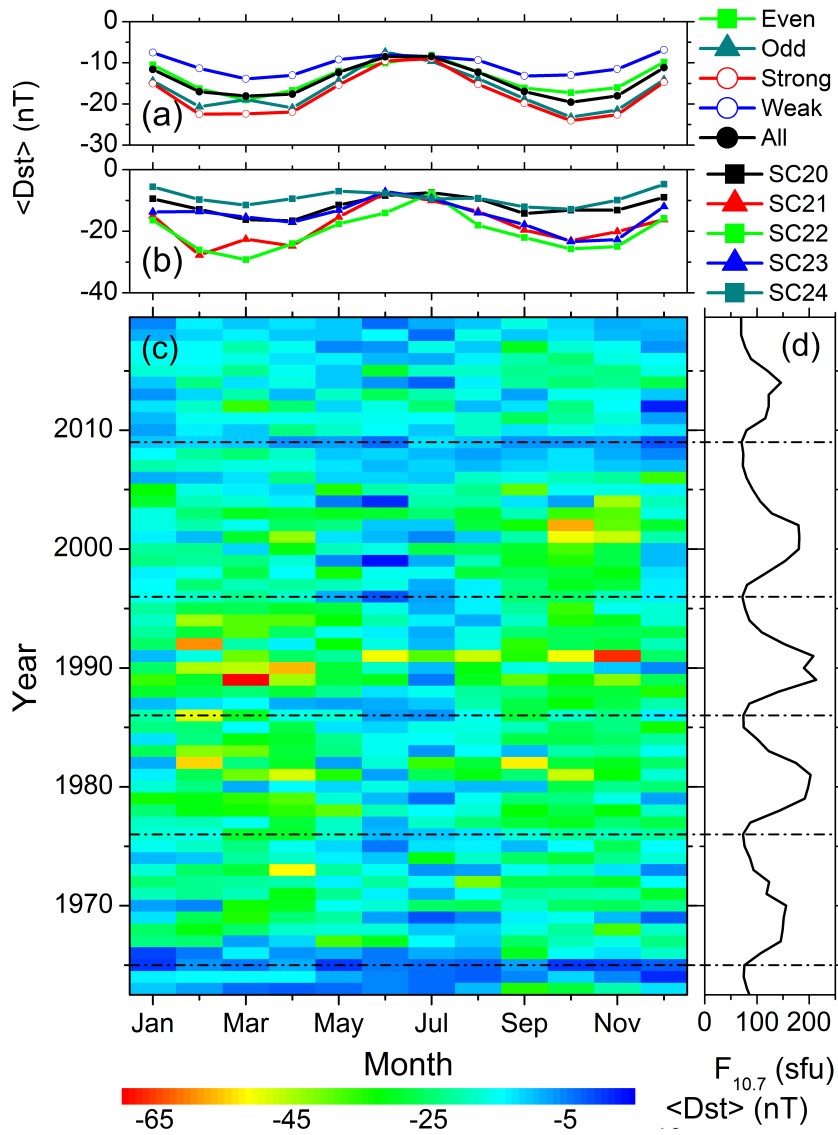

**Figure 7.** Geomagnetic Dst index variation from 1963 through 2019. Panel (c) shows the year-month contour plot of the mean Dst value in each month of the years 1963-2019. The values of different colours are given in the legend at the bottom. Data gaps are shown by crosses. Panel (d) shows the yearly mean $F_{10.7}$ solar flux. Panel (b) shows the monthly means of Dst during each solar cycles, and panel (a) shows the same during groups of the even, odd, strong, weak and all solar cycles.

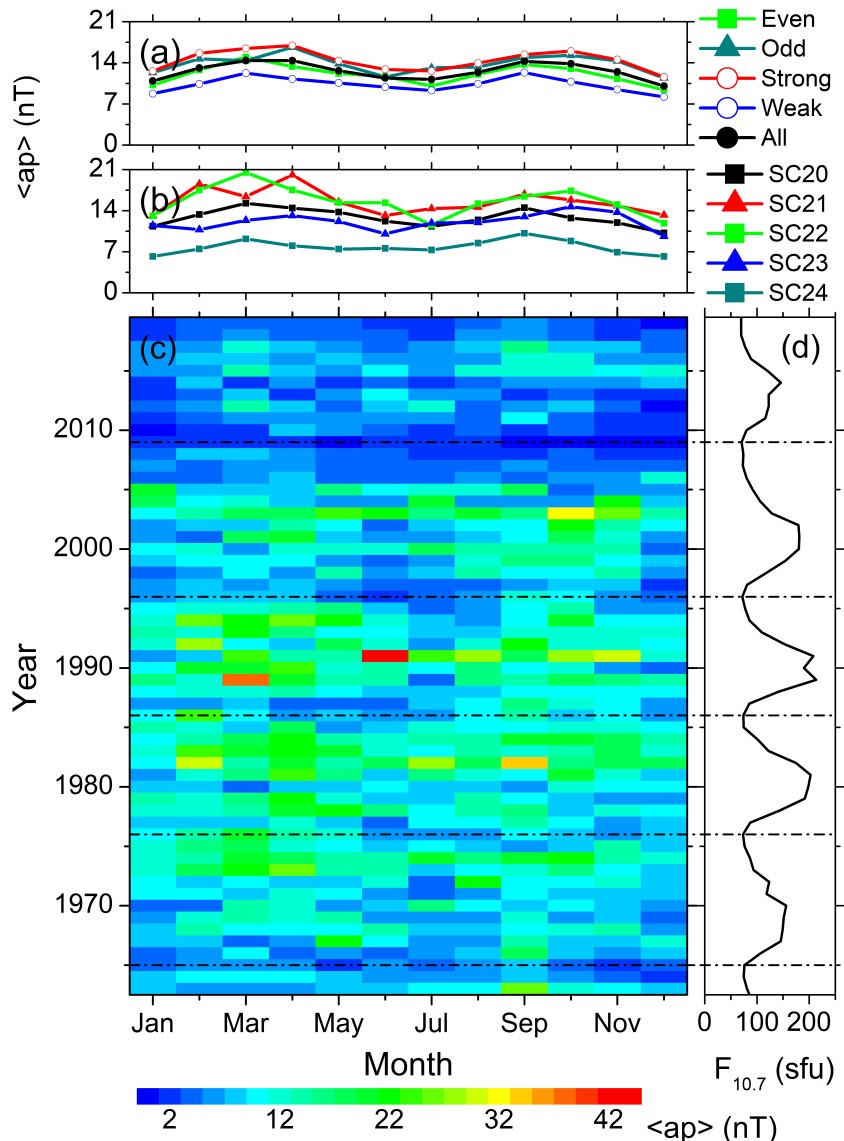

**Figure 8.** Geomagnetic ap index variation from 1963 through 2019. The panels are in the same format as in Figure 7.

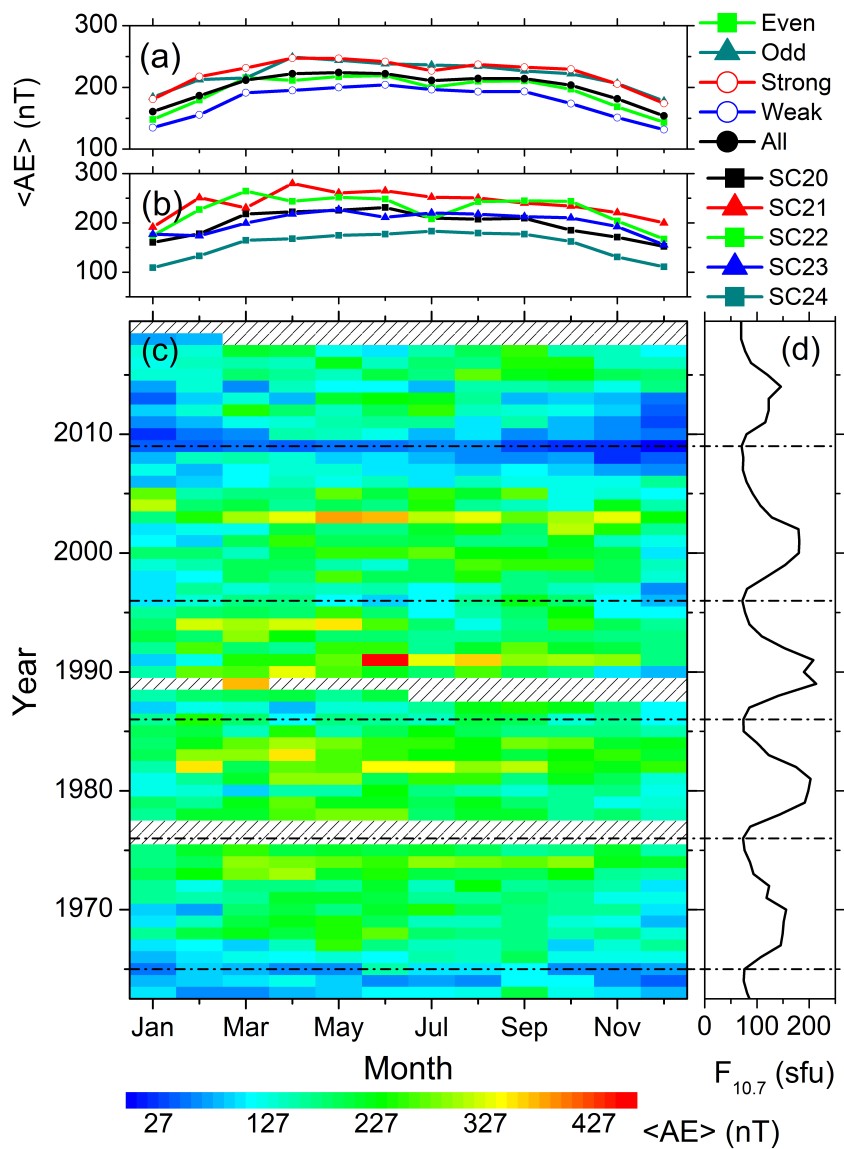

**Figure 9.** Geomagnetic AE index variation from 1963 through 2019. The panels are in the same format as in Figure 7.

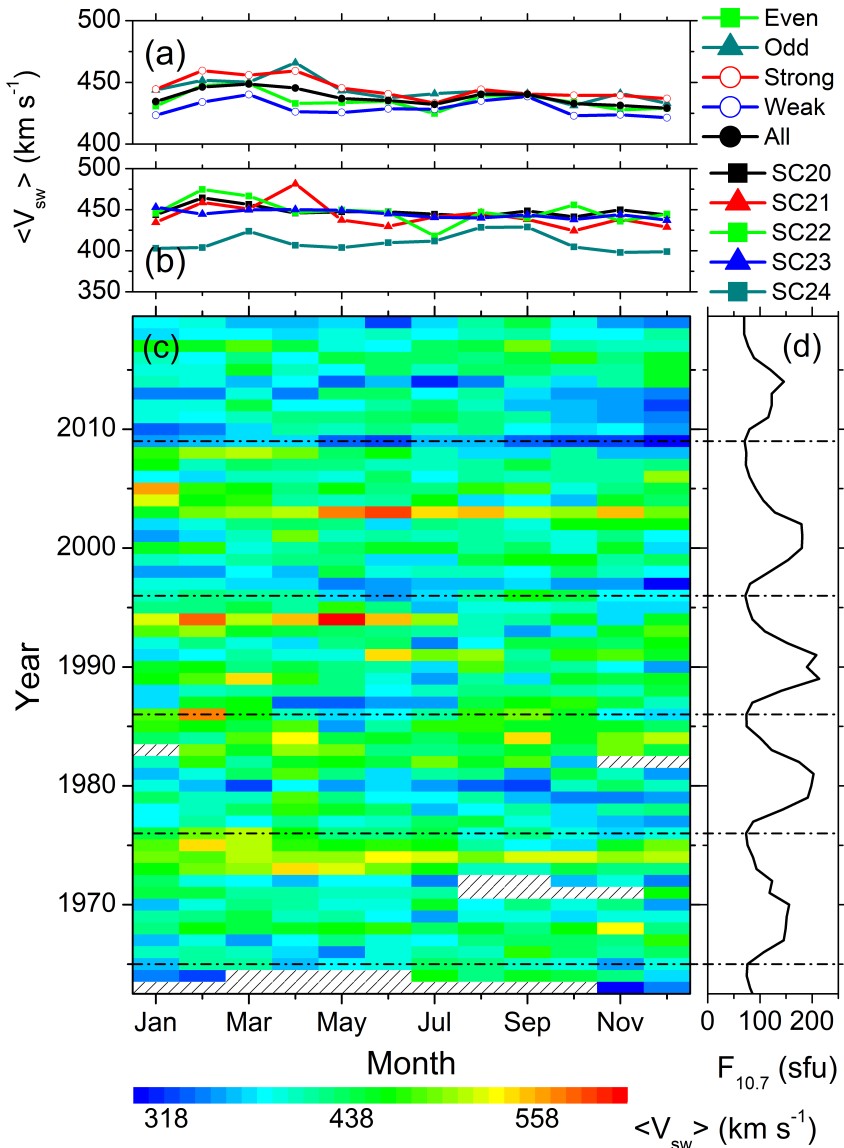

**Figure 10.** Solar wind speed $V_{sw}$ variation from 1963 through 2019. The panels are in the same format as in Figure 7.

## Solar wind-magnetosphere coupling

The periodogram analysis (Figure 3j and Table 3) identified a weak annual component in the variations of the solar wind speed $V_{sw}$ (compared with its stronger amplitude longer-scale variations). The monthly mean values of $V_{sw}$ during each year of observation are shown in Figure 10c. In a solar cycle, $V_{sw}$ peaks around the descending phase indicating a higher

occurrence rate of HSSs during this phase. This is also confirmed by the variations of $D_{500}$ (not shown). Interestingly, during the descending phase of SC20, the $V_{sw}$ peak can be noted around March-April; during the SC21 descending phase, two equinoctial peaks are almost symmetric; during the SC22 descending phase, peaks are recorded during the first half of the year; the peaks shift to the second half of the year during the SC23 descending phase; and during the SC24 descending phase, no prominent feature can be inferred. Thus, overall, a shift of the seasonal peak of $V_{sw}$ from the first half to the second half of the year can be observed between the even and the odd cycles. In addition, during the first half of the year, the average values are significantly high during the odd and strong cycles than during the even and weak cycles, respectively (Figure 10a).

Figure 11 shows the monthly mean values of the coupling function $VB_s$ during all years of observation. In a solar cycle, $VB_s$ peaks around the solar maximum, when almost symmetrical peaks can be observed during the equinoxes and minima during the solstices (Figure 11c). The lowest values of $VB_s$ are recorded during SC24 (Figure 11b). There is no prominent difference between the weak and strong cycles, and between the even and odd cycles, except that the February and October values are higher during the odd and strong cycles compared to those during the even and the weak cycles, respectively (Figure 11a).

## 4 Conclusions

We used an up-to-date database of substorms, HILDCAAs and geomagnetic storms of varying intensity along with all available geomagnetic indices during the space exploration era (i.e., after 1957) to explore the seasonal features of the geomagnetic activity and their drivers. No such study involving such a long database and all types of geomagnetic activity has been reported before. As substorms, HILDCAAs and magnetic storms of varying intensity have varying solar/interplanetary drivers, such a study is important for a complete understanding of the seasonal features of the geomagnetic response to the solar/interplanetary events. The main findings of this work are discussed below.

First, the semi-annual variation is not a "universal" feature of the geomagnetic activity. While the monthly numbers of substorms, moderate and intense magnetic storms exhibit the semi-annual variation with two equinoctial maxima and a summer solstice minimum, super storms (with a very low occurrence rate) and HILDCAA events do not exhibit any clear seasonal dependence. For geomagnetic indices, the monthly mean ring current index Dst and the global geomagnetic activity index ap exhibit the semi-annual variation, while the auroral ionospheric electrojet current index AE exhibits an annual variation with a summer solstice maximum and a winter minimum. These results clearly demonstrate varying solar, interplanetary, magnetospheric and ionospheric processes behind different geomagnetic events and indices. While the magnetic reconnection (Dungey, 1961) between the southward IMF and the northward (dayside) geomagnetic field is the key for any geomagnetic effect, variations in the reconnection process and modulation by other processes may result in different geomagnetic effects (e.g., Gonzalez et al., 1994; Tsurutani et al., 2020; Hajra, 2021a; Hajra et al., 2021, and references therein). In general, major magnetic storms are associated with strong magnetic reconnection continuing for a few hours, while weaker reconnection for an hour or less can cause substorms. On the other hand, discrete and intermittent magnetic reconnection continuing for a long interval of time may lead to HILDCAAs (see Gonzalez et al., 1994, for a detailed comparison).

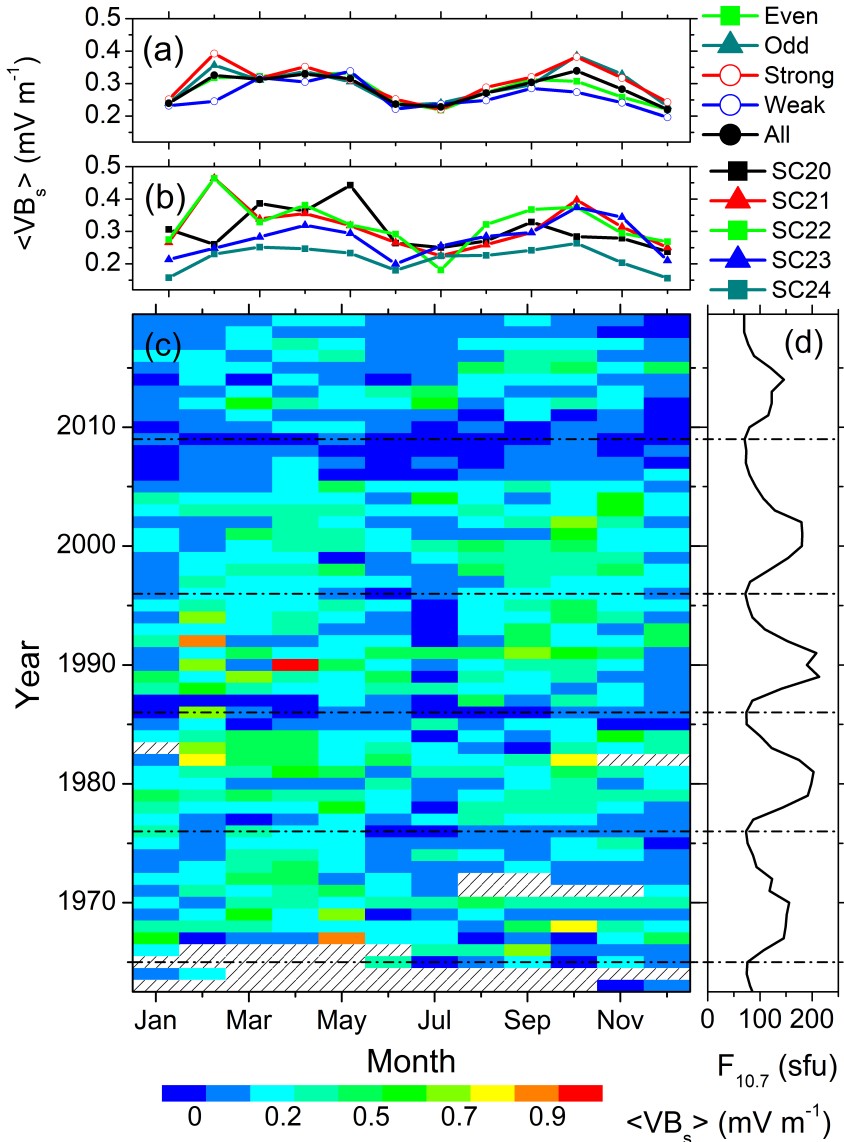

**Figure 11.** Solar wind coupling function $VB_s$ variation from 1963 through 2019. The panels are in the same format as in Figure 7.

We observe a clear semi-annual component in the coupling function $VB_s$ which represents the reconnection electric field or the magnetic flux transfer rate into the magnetosphere. On the other hand, the solar wind speed $V_{sw}$ does not have any semi-annual component, only annual and longer-scale components. As the main focus of the present work is the seasonal features, for a discussion on the longer-scale variations in $V_{sw}$, we refer the reader to previous works (e.g., Valdés-Galicia et al., 1996; El-Borie, 2002; El-Borie et al., 2020; Hajra, 2021a, c; Hajra et al., 2021, and references therein). However, this result is very interesting. This clearly implies that the solar wind does not have any intrinsic semi-annual variation, and that the semi-annual variation in $VB_s$ is due to magnetic configuration ($B_s$) as suggested previously (e.g., Cortie, 1912; McIntosh, 1959; Boller and Stolov, 1970; Russell and McPherron, 1973). The $VB_s$ semi-annual variation is suggested to cause the semi-annual variations of the substorms, the moderate and intense storms, and the geomagnetic Dst and ap indices. On the other hand, absence of any clear seasonal features in the super storms and HILDCAAs indicates more complex solar wind-magnetic coupling process during these events, which needs further study. As previously established, HILDCAAs are associated with HSSs emanated from the solar coronal holes (e.g., Tsurutani and Gonzalez, 1987; Hajra et al., 2013). Dominating longer-scale variations in $V_{sw}$ (as revealed in the present work) may be a plausible reason for the ~4.1-year variation and lack of any seasonal feature in HILDCAAs (Hajra et al., 2014a; Hajra, 2021c). Annual variation in the auroral ionospheric AE index, as mentioned before, may be attributed to a combined effect of the solar wind $V_{sw}$ variation, the asymmetric pole exposition to the solar radiation, and the ionospheric conductivity variations (see, e.g., Wang and Lühr, 2007; Tanskanen et al., 2011).

In addition to the above, we found a complex solar activity dependence of the above-mentioned seasonal features. The spring-fall asymmetry in substorms and the average $V_{sw}$ variation between the odd and even solar cycles are consistent with results reported by Mursula et al. (2011). An interesting and puzzling result is observed in terms of variations in the semi-annual variability (seasonal modulation between the equinoctial maximum and the solstice minimum) between the strong (odd) and weak (even) solar cycles. While the seasonal modulation in substorms, all storms, moderate storms and the ap index is larger during the weak (and even) solar cycles compared to the strong (and odd) solar cycles, the reverse is true for the intense storms, the Dst index and the coupling function $VB_s$. At present we do not know the exact mechanism behind this result. In fact, further study is required for a better understanding of the solar cycle dependencies of the geomagnetic activity seasonal features. In conclusion, this study, along with several previous works (e.g., Mursula et al., 2011; Hajra et al., 2013, 2016; Hajra, 2021b), calls for a careful re-analysis of the solar, interplanetary, magnetospheric and ionospheric observations before applying the theoretical semi-annual models.

*Data availability.* The solar wind plasma and IMF data used in this work are obtained from the OMNI website (https://omniweb.gsfc.nasa.gov/). The geomagnetic indices are obtained from the World Data Center for Geomagnetism, Kyoto, Japan (http://wdc.kugi.kyoto-u.ac.jp/). The list of substorms is collected from the SuperMAG website (https://supermag.jhuapl.edu/). The $F_{10.7}$ solar fluxes are obtained from the Laboratory for Atmospheric and Space Physics (LASP) Interactive Solar Irradiance Data Center (https://lasp.colorado.edu/lisird/).

*Author contributions.* RH had the original idea. AMSF and RH did the data analysis. RH prepared the first draft. All authors participated in the development, revision of the manuscript and approved the final draft.

*Competing interests.* Authors declare that there are no competing interests.

*Acknowledgements.* The work of A. M. S. F. is funded by the Brazilian CNPq agency (project no. PQ-300969/2020-1, PQ-301542/2021-0, and PQ-301969/2021-3). The work of R. H. is funded by the Science and Engineering Research Board (SERB, grant no. SB/S2/RJN-080/2018), a statutory body of the Department of Science and Technology (DST), Government of India through Ramanujan fellowship. E. E. would like to thank Brazilian agencies for research grants: CNPq (contract no. PQ-302583/2015-7, PQ-301883/2019-0) and FAPESP (2018/21657-1). The work of M. J. A. B. was supported by CNPq agency (contract no. PQ-302330/2015-1, PQ-305692/2018-6) and FAPEG
agency (contract no. 2012.1026.7000905). We would like to thank two reviewers for extremely valuable suggestions that substantially improved the manuscript.

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
