# Peer review of "Seasonal features of geomagnetic activity: a study on the solar activity dependence"

_Annales Geophysicae, 2021_

## Author Response (AR1)

**Reply to the comments by Paul PUKITE**:

We would like to thank Paul PUKITE for carefully reading the manuscript and giving valuable comments and/or suggestions. Please see below our response.

Seems like this semi-annual dependence should necessarily occur since there is a semi-annual cycle of the earth's axis declination. This would give a larger scattering cross-section to the polar regions (more susceptible to ionizing radiation) twice a year, alternating north and south pole.

- Thank you. In fact, as indicated in our manuscript, the semi-annual dependence of the geomagnetic activity is one of their earliest-reported features (e.g., Broun, 1848; Sabine, 1852). At present, there are three main mechanisms which are used to discuss this feature. The three mechanisms are:
    1. The "axial effect" proposed by Cortie (1912), which is related to the Earth's position in the heliosphere
    2. The "equinoctial effect" (Boller and Stolov, 1970), related to the relative angle of solar wind incidence with respect to Earth's rotation axis
    3. The "Russell–McPherron effect" (Russell and McPherron, 1973), related to the geometrical controls of interplanetary magnetic fields.
- These are clearly discussed in the manuscript.

**Reply to the comments by Editor**:

Dear Dr. Roussos,

We are submitting herewith the revised manuscript (# angeo-2021-27): "Seasonal features of geomagnetic activity: a study on the solar activity dependence". We thank you and the two reviewers for carefully reading the manuscript and giving valuable and constructive comments and suggestions. The manuscript is now revised based on all comments and suggestions. While the modifications are incorporated in the submitted manuscript, we are also submitting a manuscript version with "track changes" where all modifications can be clearly seen. We also submit two response files (updated) where we list all comments by the reviewers and indicate how they are incorporated in the revision.

Please see below our response to your comments and suggestions.

Thank you for submitting manuscript "Seasonal features of geomagnetic activity: evidence for solar activity dependence?" to An. Geoph. and for responding to the concerns of the two reviewers and to one external commentator in the interactive discussion.

Based on the content of the interactive discussion and my own reading of your manuscript, I share the opinion of one of the reviewers that some of the data analysis steps need reconsideration, and thus a major revision is required. I acknowledge that, based on your responses in the interactive discussion, that you are confident that the concerns of the 1st reviewer, when taken into account, they have no impact on your original results. Still, I would be keen to see the corresponding version of the revised article that is in line with your responses provided in the discussion.
- Thank you for the suggestion. We have now updated the response to the reviewer #1 (and #2). We assure that all the suggestions are taken into account in the revision now. These are now indicated in the response files as well as in the manuscript version with "track changes".

Note also that Reviewer #1 provided a series of comments in an Annotated PDF to which I could not find any responses from you. Since, however, the major comments from the annotated manuscript were covered also elsewhere, we can proceed with the next stages of peer review.
- We are sorry that we did not include our response to the comments annotated in the PDF. Now we include them in our response file (updated). In addition, we assure that all of the comments/suggestions are now incorporated in the revised manuscript.

For that purpose, please provide us, along with the revised manuscript, a formal response to the two reviewer comments (including those in the annotated manuscript), indicating where changes have been done with respect to the original submission. Some of the responses may be copied from the interactive discussion, if you think they require no update. I anticipate that your revised version would be sent back to at least one of the two original reviewers for a second round of reviews.
- Yes, as you suggested, we now submit (i) the revised manuscript where all corrections are incorporated, (ii) the revised manuscript version with "track changes" where the

modifications can be clearly identified, (iii) response files where all comments (including those in the annotated PDF file) are listed followed by our response on how they are incorporated in the revision.

With best regards,

Adriane Marques de Souza Franco

**Reply to the comments by Anonymous Referee #1**:

We would like to thank the Referee #1 for carefully reading the manuscript and giving valuable comments and suggestions. We greatly appreciate your detail markups in the original manuscript (listed separately at the end and answered). All your comments and suggestions are now incorporated in the revised manuscript.

I have read the manuscript "Seasonal features of geomagnetic activity: evidence for solar activity dependence?". The authors present an extended analysis of the semi-annual variation occurrence in various solar wind parameters, geomagnetic indices and the occurrence rates of storms with various magnitudes, substorms and HILDCAAs. Nevertheless, there are points in the manuscript that need further clarification and, moreover, there are certain aspects of the statistical analysis which need further testing. Therefore, my suggestion is major revision.
-        Thank you.

Even though all my comments are included in the attached pdf, I'm pointing out some important comments below.
-        Thank you. We have carefully gone through your comments/suggestions included in the annotated pdf (listed below and responded), and incorporated all of them in the revised manuscript.

1) Even though it is not adequately explained, the reader understands that the authors use the monthly mean of the occurrence rate of substorms, HILDCAAs, etc. to perform statistics. If indeed the authors are using monthly mean of the occurrence, it could introduce several artifacts in the results due to very low values. For example, in figure 1, the occurrence of HILDCAAs or super storms take only a couple of values (0, 1, 2). It would be helpful to provide the same results using the total occurrence rate per month instead of the mean. Another option would be to normalize the monthly occurrence rates with respect to the maximum occurrence for the whole dataset.
-        Thank you for the comment. We are sorry for the confusion. In fact, we use the monthly means of $F_{10.7}$, Dst, ap, AE, $B_0$, $V_{sw}$, $D_{500}$, $VB_s$ and ε, and the monthly numbers of substorms, HILDCAAs and magnetic storms of varying intensity. This is now made clear in the revised manuscript (lines 133-137 and 186-187 of the revised manuscript with "track changes").

2) The significance level in the Lomb periodogram, as a statistical metric, is much affected by the strongest periodicity (e.g. 11 years). This could result to artifacts when discussing much lower periodicities which statistically be weaker and probably showed below this confidence level. One way to overcome this feature is to filter the time-series in the desired period range (either way the 11 years periodicity is well known and of no importance for the present work). Another way is to limit the Lomb periodogram in the desired range (for example 3 - 24 months).
-        Thank you very much for the suggestion. We now show the periodograms based on the original database of 1 month resolution (see above), as well as the periodograms after filtering out the dominating ~11-year periodicity from the data (Figure 3).

3) The authors should further discuss the reason why the occurrence of substorms exhibits the semi-annual variation, while the AE index, which is a proxy for substorm activity, does not.

- This is now discussed in the text, as suggested (lines 161-176).

4) The authors should discuss the discrepancies between odd/even and strong/weak cycles after they have clearly stated what a strong/weak cycle is.

- Thank you for the comment. The strong/weak cycles are defined in section 2 (liners 113-120), and the discrepancies are now discussed (Table 4, lines 238-250, 347-358).

Finally, I think that the question mark in the title of the manuscript is contradicting. If the conclusions of this work are indeed correct, then there is a dependence in Solar activity.

- Thank you. The question mark is now removed and title of the manuscript is modified.

We list below your comments/suggestions in the annotated PDF file followed by our response:

1. Line 19: currents → current
- Done (line 20).

2. Line 19: enhancement → enhancements
- Done (line 19).

3. Line 21: "Storms can continue for a few hours to a day." → "Storms duration spans a few hours to several days."
- Done (lines 21-22).

4. Line 21: an hour → a few hours
- Done (line 23).

5. Line 22: "disturbances in the auroral region" → This is not correct. Substorm activity is the term used for a series of processes taking place at the nightside magnetosphere including dipolarization and particle injection. Maybe you are referring to the results of substorms in the auroral region. Please rephrase.
- Thank you very much. The statement is now corrected (lines 23-26).

6. Line 27: rarer → less
- Done (line 48).

7. Line 28: variations are → variation is
- Done (line 49).

8. Lines 28 – 34: "The semi-annual variations are reported in the occurrence rates and intensities of the magnetic storms …See Lockwood et al. (2020) for an excellent discussion of the mechanisms." → The authors do not sufficiently report the various mechanisms that are responsible for the semi-annual variation.

For example, Cliver et al. 2000 and O'Brien and McPherron, 2002 showed that the semi-annual variation in geomagnetic indices AE, Dst and am was mostly due to the equinoctial effect, while Kanekal et al 2010 suggested that the semi-annual variation in the relativistic electron fluxes of the outer belt was a result of the Russell-McPherron effect. This was recently proven by Katsavrias et al. 2021.

Katsavrias et al. 2021, Ann. Geophys., 39, 413–425, https://doi.org/ 10.5194/angeo-39-413-2021Cliver, E. W., Kamide, Y., and Ling, A. G.: Mountains vs. valleys: the semiannual variation of geomagnetic activity, J. Geophys. Res., 105, 2413–2424, https://doi.org/10.1029/1999JA900439, 2000.

Kanekal, S. G., Baker, D. N., and McPherron, R. L.: On the seasonal dependence of relativistic electron fluxes, Ann. Geophys., 28, 1101–1106, https://doi.org/10.5194/angeo-28-1101-2010, 2010. O'Brien, T. P., and R. L. McPherron, Seasonal and diurnal variation of Dst dynamics, J. Geophys. Res., 107(A11), 1341, doi:10.1029/2002JA009435, 2002.

- Thank you very much for the suggestion and the references. We now elaborated our discussion with help of the suggested references (lines 55-58).

9. Line 54: peak → Typically it is Dst min. Please change it everywhere in the text.
- Done (line 80, and several others).

10. Lines 63 – 64: "While substorms occur during HILDCAAs, they represent different magnetosphere/ionosphere processes (Tsurutani et al., 2004; Guarnieri, 2005)." → It would be helpful for the reader if the authors dedicated a couple of lines explaining the differences between substorms and HILDCAAs from a physical point of view.
- As suggested, we now elaborated the differences between substorms and HILDCAAs (lines 29-40).

11. Line 71: "to the percentage of days" → The percentage compared to what? A day, a week?
- The definition is now made clear as: "The $D_{500}$ parameter is defined as the percentage of days with the peak solar wind speed $V_{sw}$ equal or higher than 500 km s$^{-1}$ in each month of a year" (lines 99-100).

12. Line 72: "This parameter indicates the occurrence of the solar wind high-speed streams (HSSs)." → How can this be true? ICMEs can also have Vsw > 500 km/s.
- The statement is now deleted.

13. Line 75: "VBs is also called the reconnection electric field." → I'm not familiar with this term so please provide reference. I know that VBs is the Half-wave rectifier introduced by Burton, R. K., McPherron, R. L., & Russell, C. T. (1975). An empirical relationship between interplanetary conditions and Dst. Journal of Geophysical Research, 80(31), 4204–4214. https://doi.org/10.1029/JA080i031p04204
- References are now provided (line 102).

14. Lines 95 – 96: "Figure 1 shows the variations of the monthly mean solar F10.7 flux, HILDCAAs and substorms, magnetic storms of varying intensity," → How is the monthly mean of substorms calculated? Wouldn't it make more sense to use the total number of substorms during a specific month?

    If indeed the authors are using monthly mean of the occurrence, it could introduce several artifacts in the results due to very low values. For example, in figure 1, the occurrence of HILDCAAs or super storms take only a couple of values (0, 1, 2).
    - Thank you for the comment. We, in fact, used the "monthly numbers" of HILDCAAs, substorms and magnetic storms. This is now clearly stated in the revised manuscript (lines 133-137 and 186-187).

15. Lines 96 – 97: "percentage occurrences of Vsw ⩾ 500 km s−1 (D500)," → Again how is this calculated?
    - It is now defined. Please see our response above.

16. Line 98: "there are several short-term fluctuations in the data" → I think that this is a quite poor description of the figure 1. Please provide an adequate description of the results.
    - As suggested, we now extended and improved the description of Figure 1 (lines 133-144).

17. Line 102: normalized → This is not normalization. This is simply the mean occurrence rate for each month.
    - Corrected (line 147).

18. Lines 105 – 107: "Substorm occurrence rate clearly exhibits two peaks during the months of March and October, and a summer solstice minimum (during the month of June). HILDCAAs do not exhibit any clear seasonal feature, except a significant minimum in November. Geomagnetic storms, from moderate to intense, exhibit a clear semi-annual variation." → It would be helpful to provide the same figure using the total occurrence rate per month instead of the mean. Especially for HILDCAAs and super storms, which are rarer phenomena, you have very low values (occurrence) that can affect your statistics.

    Another option would be to normalize the monthly occurrence rates with respect to the maximum occurrence for the whole dataset.
    - Thank you for the suggestion. As explained above, we are using the total number of events in each month. This is now made clear in the revision (lines 133-137).

19. Figure 2 caption: Please include units in the figure labels.
    - Done.

20. Line 120: "This can be done in a future work." → I believe that writing a whole new paper just for AL and AU cannot stand. Moreover, it has been already shown by

previous works, that Since it is a relatively easy work I would suggest to include AL index in this manuscript.

In a previous study by Katsavrias et al. 2016 (https://doi.org/10.1016/j.asr.2016.03.001) was indicated that there is no semi-annual variation in AE index. On the other hand, Lockwood et al, 2020 (https://doi.org/10.1051/swsc/2020023) showed that the semi-annual variation is indeed present in the AL index. If the AE results are different you should at least suggest why.

- As suggested by you (and the other reviewer), this part is now elaborated with additional references and discussion of previous works (lines 161-176).

21. Line 128: "Lomb-Scargle periodogram analysis" → What is the resolution of the dataset? Is the periodogram applied to monthly data? Once again you should be very careful with the magnitude of the occurrence rates. If HILDCAAs take values in the 0-2 range, it could affect the significance level of the periodogram.
- Thank you very much for the comment. We now clearly mention that "we use the monthly means of $F_{10.7}$, Dst, ap, AE, $B_0$, $V_{sw}$, $D_{500}$, $VB_s$ and ε, and monthly numbers of substorms, HILDCAAs and magnetic storms of varying intensity" (lines 186-187).

22. Lines 129 – 133: "As expected, the $F_{10.7}$ solar flux shows a prominent…However, we are interested in annual or shorter-scale periodicities in the events and parameters." → There are some features here I would like to mention.

As mentioned before, the authors perform the Lomb periodogram without informing us for the resolution of the dataset used. First, judging from the lowest period value (2 months) I guess they use monthly resolution. Moreover, they expand the periodogram up to 30 years indicating the strong 11 -years periodicity.

The significance level, as a statistical metric, is much affected by the strongest periodicity (e.g. 11 years). This could result to artifacts when discussing much lower periodicities which statistically be weaker and probably showed below this confidence level.

One way to overcome this feature is to filter the time-series in the desired period range (either way the 11 years periodicity is well known and of no importance for the present work). Another way is to limit the Lomb periodogram in the desired range (for example 3 - 24 months).
- Thank you very much for the comments. As mentioned above, we now clearly mention the resolution of the data. Based on your suggestion, we are now showing the LS periodogram of the original data, and LS periodogram of the data after filtering out the 11-year periodicity from the data (Figure 3).

23. Figure 3 caption: "and solar wind parameters IMF $B_0$, $V_{sw}$, $D_{500}$, $VB_s$ and ε-parameter in the same panel," → Last panel is really hard to read (for example I cannot distinguish the Vsw results). Please separate the parameters into two panels.

- Done.

24. Lines 162 – 165: "The bottom right panel shows the yearly mean $F_{10.7}$ solar flux…the even, odd, strong, weak and all solar cycles." → It is really hard for the reader to understand which panel is mentioned each time. Please label the panels with letters (a, b, c, etc.) and repeat it for every figure in the manuscript.
- Done.

25. Table 3 caption: "the weak and strong solar cycles." → Please define weak and strong solar cycle.
- They are now defined in the Data section (lines 113-114).

26. Line 245: while
- Added (line 327).

27. Lines 254 – 255: "This has a large contribution in the semi-annual variations of the substorms, moderate and intense storms, and geomagnetic Dst and ap indices." → The authors should further discuss the reason why the occurrence of substorms exhibits the semi-annual variation, while the AE index, which is a proxy for substorm activity, does not.
- As suggested, the discussion is now elaborated in the revised manuscript (lines 161-176).

**Reply to the comments by Anonymous Referee #2**:

We would like to thank the Referee #2 for carefully reading the manuscript and giving valuable comments and suggestions. All your comments and suggestions are now incorporated in the revised manuscript.

**Summarization**:

The work intends to study the Earth's geomagnetic seasonal features produced by solar influences. Concerning the main geomagnetic activities, the authors consider the Sun-Earth electrodynamics coupling modulation captured by (1) geomagnetic indices: the low-latitude equatorial effects (Dst), planetary effects ap, auroral effects AE, and Akasofu Epsilon parameter, and (2) the solar influences defined by some significant parameters: the solar flux (the F10.7), Interplanetary magnetic field Magnitude Bo, the South-North oriented component (supposed to be in GSM) Bs, the solar wind speed Vsw, the D500 (percentage of the days with the Vsw peak equal or higher than 500 km/s). Although several periods have been obtained by signal analysis from the Lomb-Scargle periodograms, the investigation focus is short cycles, less or equal to an annual variation.

The data interval involves the solar cycles 20 to 24.

**General comments**:

The work is interesting. The text is well written using clear ideas. The contents are well enchained. The statistical technique is simple. Nevertheless, the current work needs some improvements. Beyond the simple statistical interpretation, some physical discussions are expected to be included.
   - Thank you for the comments. We now revise the manuscript based on all your suggestions and/or comments.

**Suggestions (major remarks)**:

In the introduction, the authors can describe the supposed sources of the Earth Geomagnetic activity modulations more clearly.

On the one hand: the solar cycle activity (11 year), the solar rotation (27 days), the solar activity features (in general lines): the electromagnetic radiation, the corpuscular radiation, plasmas emission phenomena, the heliospheric current region occurrence. On the other hand: the Earth's translational movement (solstices), the inter-hemispheric symmetry (equinoxes), and the effect of the Coordination reference systems (GSE x GSM, for instance).
   - Done. We now describe more clearly the plausible sources of the Earth's geomagnetic activity modulations as suggested by you (lines 41-46 of the revised manuscript version with "track changes").

**Discussion problems not adequately addressed**:

**Line 115**: "...Thus the AE index shows an annual variation...". There are several geomagnetic stations located only in the North Hemisphere to calculate AE (http://wdc.kugi.kyoto-u.ac.jp/aedir/ae2/AETABLE1.html). The asymmetric pole exposition to the Sun during the Earth translation could contribute to this annual variation.
- Thank you for the suggestion. We now include a discussion of asymmetry in AE observatories that can contribute to the AE annual variation (lines 162-165).

**Line 121**: "...VBs exhibits a semi-annual variation...". It was not declared at work; however, the GSM seems the choice (from the OMNI web service option), which creates a (statistical) artefact. The Bs calculated is affected by a diary cycle (magnetic dipole attitude spinning around the rotation axis) and by a 6-month interval, i.e., this latter concerns the attitude of the Earth rotation-axis during the translation.
- Thank you for the comment. Yes, the GSM coordinates are used, we have added a description about it in section 2 (lines 126-129). We now consider the possible impact from the magnetic dipole attitude spinning on the Bs variation, as suggested.

**Line 136**: "...HILDCAAs, on the other hand, exhibit a ~4.1-year periodicity...". Could it be related to the 11-year solar activity cycle (concerning the ascending and descending phases)?
- HILDCAAs generally occur more frequently during the descending phase of a solar cycle, when Earth is more frequently impacted by solar wind high-speed streams (HSSs) emanated from solar equatorial coronal holes. Solar wind speed Vsw is found to exhibit several periodicities including a ~4.7-year periodicity. Thus, the ~4.1-periodicity seems to be associated with the Vsw variation. This is now made clearer in the revised manuscript (lines 218-219).

**Line 151**: "...On the other hand, the ~1-year periodicity in Vsw/D500 can be a source of the annual variation in the AE index...". Beyond the Line 121 remark, could the Earth's slightly elliptical orbit also contribute to it?
- Thank you for the comment. However, our thought is that the Earth's elliptical orbit contribution is very small particularly, for a long-term study, such as the present one. At present, the exact contribution is not known (so cannot be addressed in the paper), but should be investigated.

**Lines 229-270**: The Conclusion section will be affected by the earlier remarks. Please, pay attention to other parts of the text. Some written contents in Results Section will require an update of interpretation or discussion.
- Thank you. Done.

**Minor remarks**:

**Line 86-87**: Please, justify the solar cycles to be grouped into the even and odd ones.
- We now discuss the differences in the "even" and "odd" solar cycles as a "consequence of the nonlinear interactions that provide the stabilizing mechanism for the cycle's amplitude" (Durney, Solar Physics 2000) (lines 118-120).

**Figure 3**: The bottom panel presents colour (VBs and D500) confusingly.
- The figure is now improved.

**Line 146**: write "...in the southern hemisphere..."
- Done (line 211).

---

## Author Response (AR2)

**To the Editor**:

Dear Dr. Roussos,

We are submitting herewith the revised manuscript (angeo-2021-27): "Seasonal features of geomagnetic activity: a study on the solar activity dependence". We have now incorporated all minor comments by Referee #1 in the revised manuscript.

Kind regards,

Adriane Marques de Souza Franco

**Reply to the comments by Referee #1**:

Thank you again for carefully reading the revised manuscript and giving valuable suggestions. The manuscript is now corrected based on all your minor comments/corrections. The modifications are clearly marked the revised manuscript with "track changes".

I have read the revised manuscript "angeo-2021-27" and I have to say that the authors have adequately responded to all my comments and, moreover, have successfully modified the manuscript. Even though the manuscript has been substantially improved I have 3 more minor comments/corrections which should be implemented.
- Thank you.

1) In lines 175-176 of the revised manuscript, the authors state: "may indicate that AE is dominated by the eastward ionospheric current (AU) rather than the substorm related westward current (AL)". At first sight this is wrong. Since the westward current is much more intense (AL values are usually much greater than the ones of AU) the AE index is definitely dominated by the AL index. Nevertheless, when it comes to certain frequency components, the contribution is relative. The authors should discuss about the semi-annual component only and not about the general contribution to AE.
- Thank you for the suggestion. The misleading statement is now removed.

2) In line 181, the authors state: "No clear seasonal features can be inferred from the variations of the monthly mean Vsw (Figure 2k, legend on the left), D500 (Figure 2k, legend on the right) ....". I strongly disagree. There are clear peaks on March and September in D500. Please modify accordingly.
- Thank you for pointing out the error. The statement is now corrected (lines 169 – 171 in the revised manuscript with "track changes").

3) In lines 201-202, the authors state: "HILDCAAs (Figure 3d), on the other hand, exhibit a ~4.1-year periodicity, while no annual or lower-scale variation was recorded". There is still the issue of HILDCAAs values as I mentioned in my previous review. Since the values are mostly 0, 1 and 2, the variation of the specific time-series is negligible. This may introduce several artifacts to the spectral analysis. Thus, I would strongly suggest that the authors discuss this issue at this point, indicating that the results of the LS periodogram for

HILDCAAs cannot be fully trusted due to the aforementioned limitation. The same applies for superstorms as well.

- Thank you. We now discussed the issues of HILDCAAs and superstorms, as suggested (lines 190 – 193).